# Rapid and Green Classification Method of Bacteria Using Machine Learning and NIR Spectroscopy

**DOI:** 10.3390/s23177336

**Published:** 2023-08-23

**Authors:** Leovergildo R. Farias, João dos S. Panero, Jordana S. P. Riss, Ana P. F. Correa, Marcos J. S. Vital, Francisco dos S. Panero

**Affiliations:** 1Instituto Federal de Roraima, Campus Boa Vista, Av. Glaycon de Paiva, 2496 Pricumã, Boa Vista 69303-340, Brazil; leovergildofarias@yahoo.com.br (L.R.F.); joao.panero@ifrr.edu.br (J.d.S.P.); 2Instituto Federal de Roraima, Campus Novo Paraíso, BR-174, Km-512—Vila Novo Paraíso, Caracaraí 69365-000, Brazil; jordana.riss83@gmail.com; 3Postgraduate Program in Natural Resources-PRONAT, Universidade Federal de Roraima, Av. Cap. Ene Garcês, 2413-Aeroporto, Boa Vista 69310-000, Brazil; folmercorrea@gmail.com (A.P.F.C.); marcos.vital@ufrr.br (M.J.S.V.)

**Keywords:** bacteria, green chemistry, machine learning, near infrared

## Abstract

Green Chemistry is a vital and crucial instrument in achieving pollution control, and it plays an important role in helping society reach the Sustainable Development Goals (SDGs). NIR (near-infrared spectroscopy) has been utilized as an alternate technique for molecular identification, making the process faster and less expensive. Near-infrared diffuse reflectance spectroscopy and Machine Learning (ML) algorithms were utilized in this study to construct identification and classification models of bacteria such as *Escherichia coli*, *Salmonella enteritidis*, *Enterococcus faecalis* and *Listeria monocytogenes*. Furthermore, divide these bacteria into Gram-negative and Gram-positive groups. The green and quick approach was created by combining NIR spectroscopy with a diffuse reflectance accessory. Using infrared spectral data and ML techniques such as principal component analysis (PCA), hierarchical cluster analysis (HCA) and K-Nearest Neighbor (KNN), It was feasible to accomplish the identification and classification of four bacteria and classify these bacteria into two groups: Gram-positive and Gram-negative, with 100% accuracy. We may conclude that our study has a high potential for bacterial identification and classification, as well as being consistent with global policies of sustainable development and green analytical chemistry.

## 1. Introduction

In 1998, Paul Anastas and John Warner proposed the terminology Green Chemistry. Chemistry itself began to change, and in this same period, a branch of Chemistry called Green Chemistry was born. This branch would play a fundamental role in this new policy of sustainable development, mainly concerning pollution prevention, through the reduction or elimination of laboratory waste and the use or generation of substances that are dangerous to the environment and the health of the population. The same authors proposed 12 principles that would guide Green Chemistry: one—Prevention; two—Atom Economy; three—Less Hazardous Chemical Syntheses; four—Designing Safer Chemicals; five—Safer Solvents and Auxiliaries; six—Design for Energy Efficiency; seven—Use of Renewable Feedstocks; eight—Reduction of Derivatives; nine—Catalysis; 10—Design for Degradation; eleven—Real-Time Analysis for Pollution Prevention and twelve—Inherently Safer Chemistry for Accident Prevention. These principles had the function of maximizing the efficiency of human and financial resources in order to minimize environmental and human health risks [1,2,3].

The most important principle is the first (because avoiding waste is better than treating or cleaning), the waste generated; the task of scientists, especially analytical chemists, is to develop new methods or techniques that can reduce or eliminate the consumption of reagents and minimize the work time of analysts through more efficient and effective routine analyzes because of the sustainable development of society [2]

In 2015, the United Nations General Assembly created the 2030 Agenda for Sustainable Development, in which it established 17 sustainable development goals broken down into 169 goals to be achieved by the year 2030. Researchers in the field of Chemistry play a role important for Sustainable Development Goals Three, Six and Nine. It is important to emphasize that the work of the chemical scientific community to the Sustainable Development Goal Nine: Industries, Innovation and Infrastructure, a greater contribution and investment in the amount of research and innovation in chemistry is needed, making researchers contribute significantly by developing new processes and methods, which update and modernize the infrastructure of industries to become more sustainable [4,5,6]. Pollution prevention involves the development of analytical methods and techniques to avoid the generation of waste, thus eliminating the need for further treatment or disposal. 

Infrared spectroscopy can meet all these criteria mentioned above: Green Chemistry, Sustainable Development Goals and White Analytical Chemistry; it has been replacing traditional analysis techniques due to its accuracy and speed in obtaining the results [7].

The region of the spectrum corresponding to the infrared is located after the visible region and covers radiation with wavenumbers in the range of approximately 12,800 to 10 cm^−1^. This region is basically divided into three bands: the near-infrared—NIR (12,800 to 4000 cm^−1^), Middle Infrared—MIR (4000 to 200 cm^−1^) and Far Infrared—FIR (200 to 10 cm^−1^). There is no specific limit defining where one region ends, and another begins; it is common to find different values for each range in the literature [8,9].

Several works have been published in the literature using near-infrared spectroscopy and the use of Machine Learning to predict chemical attributes of grains and classify rice, soy, sesame, edible oils, plants and Amazonian oils [2,7,8,9,10,11,12,13]. Furthermore, it stands out for not requiring any preparation of the samples besides not using reagents, and it does not destroy or modify the sample.

NIR spectroscopy uses different types of measurements, which will depend on the type of sample (solid, liquid, gaseous), whether it is in pasty, granulated, powdered form, etc. [8,9,10]. Recently, our research group published an article using multivariate analysis techniques, chemometrics and Machine Learning algorithms (HCA—Hierarchical Cluster Analysis, PCA—Principal Component Analysis, KNN—K-Nearest Neighbor and SIMCA—Soft independent modeling of class analogy) to classify sesame cultivars and strains, quickly, non-destructively and effectively using spectroscopy in the near-infrared, replacing morphological and molecular analysis markers, given that these techniques require different types of reagents and experienced analysts. In order to preserve the environment, we call this “NIR genotyping” [11].

Due to these excellent results, we are working on the application of NIR spectroscopy in microbiology as the production of waste from health services, mainly laboratory waste from Microbiology and Clinical Analysis Laboratories, is a social problem that must be considered in view of the environmental crisis we are experiencing today. During the pre-analytical and analytical phase, solid waste and liquid effluents are generated, which can pose risks to analysts as well as pollute the environment if these wastes are not properly treated and disposed of as much as there are national and international laws that emphasize the need to minimize waste generation and reduce risks to the environment. It is necessary to go one step beyond legislation to develop analytical techniques capable of not generating waste and effluents in these procedures.

In the last decade, NIR spectroscopy has been applied to identify bacteria, fungi and viruses, combined with Machine Learning algorithms and Multivariate Analysis techniques [14,15,16,17,18].

The identification and characterization methods of bacteria have lengthy procedures to obtain results, whether biochemical or molecular, require professionals with technical qualifications, and some methods can take weeks (Considering the steps of extraction, amplification, purification and DNA sequencing).

Bacteria are defined as prokaryotic microorganisms, which are unicellular organisms that do not have an endoplasmic reticulum, nuclear membrane, Golgi complex and mitochondria, whose reproduction occurs via asexual division [19]. Since the 1980s, various chemical, biochemical and colorimetric tests and methods have been used for the differentiation of bacteria. Tests with lactose, indole, sucrose, citrate, malonate, mannitol, hydrolysis, catalase, oxidase, gas production and among others.

To analyze the differences in bacteria, a method used is the Gram staining test, which consists of differentiating Gram-positive and Gram-negative bacteria. This method differentiates bacteria based on differences in cell walls. Gram-positive, which includes bacteria that have an extensive peptidoglycan cell wall and are stained purple by crystal violet after the laboratory technique, and Gram-negative, which have a polysaccharide outer membrane exclusive to this type of bacteria. This latter class of bacteria loses the crystal violet-iodine complex during discoloration with alcohol washing but retains safranin and appears reddish or pink after laboratory tests. The Gram staining method was named after the pathologist Hans Christian Joachim Gram, who introduced this methodology in 1884 and, to this day, remains the most used method in clinical analysis and microbiology laboratories [20,21,22].

Fuchsin presents risks regarding tissue neoplasia due to intoxication by occupational activities and/or environmental contamination, being considered carcinogenic to humans, in addition to being one of the rare dyes that have the property of becoming flammable under certain conditions. Several researches have been developed to degrade this pollutant due to the high risk of deliberate disposal of this type of dye in bodies of water, aquifers and supply sources [23].

Scientists have been fulfilling their social and sustainable responsibility through the development of new techniques and methodologies for cleaner, faster and more sustainable analysis.

The objective of this research was to apply NIR spectroscopy and Machine Learning to identify and authenticate: *Escherichia coli*; *Salmonella enteritidis*; *Enterococcus faecalis*; *Listeria monocytogenes*. And classify these bacteria into Gram-negative and Gram-positive.

## 2. Materials and Methods

### 2.1. Reactivation of Bacterial Standards

In order to verify the reliability of the bacteria identification process using near-infrared spectroscopy, for bacteria, standard (ATCC) Gram-positive strains were used: *Listeria monocytogenes* (ATCC 7644) and *Enterococcus faecalis* (ATCC00531). Gram-negative: *Salmonella enteritidis* (ATCC 13076) and *Escherichia coli* (ATCC 10536). These strains were obtained from the collection of the Federal University of Roraima. These strains were reactivated in brain heart infusion (BHI) broth, and the culture was then incubated at 35 °C for 24 h.

### 2.2. Near-Infrared Measurements

The NIR spectra were obtained through the BOMEM MD-1600 spectrophotometer (ABB Bomem, Zürich, Switzerland), equipped with a diffuse reflectance accessory (ABB Ltd., Zurich, Switzerland), with a resolution of 4 cm ^−1^, the spectral region from 14,000 to 4000 cm^−1^ or 714–2500 nm and 50 scans (co-added). The spectrum of a polytetrafluoroethylene-PTFE sample was used as background; the bacterial colonies were placed on slides and covered with a coverslip, then added on the reflectance accessory, and on the samples, the diffuse reflectance pattern (Figure 1) so that there were no errors in the readings, or entry of radiation from the laboratory environment.

### 2.3. Multivariate Analysis and Machine Learning 

The Hierarchical Cluster Analysis—HCA and the Principal Component Analysis—PCA are considered unsupervised pattern recognition (PR) methods that are used to examine the similarities or differences between samples [8,24,25,26].

HCA is a cluster analysis method, hierarchical and agglomerative, in which the samples are grouped together (clusters) in a hierarchical way from the closest (similar) to the most distant, then expressed in a tree structure or dendrogram; several algorithms are used to define the proximity between samples and/or groups, such as [24,25,26,27,28].

PCA is used to maximize the information that can be extracted from a set of spectroscopic data, as it correlatively transforms an original set of variables into a smaller set of variables that contain most of the information from the original set. Reduces the dimensionality of the original data by grouping the correlated variables, producing a new set of variables called principal components (PCs). These PCs are built like linear combinations of the original variables. The first new axis, PC1, represents the maximum variation; PC2 is orthogonal PC1, generating the graphs of scores and loadings [24,25,26].

Machine Learning—ML, is a subset of Artificial Intelligence—AI. K-Nearest Neighbor—KNN is an ML algorithm for classification, supervised, non-parametric, discriminating and deterministic pattern recognition method. The algorithm computes the similarity and finds the k the closest training examples in the dataset using the distance function; for the K number of the nearest neighbors, the distance between the query examples and all the training cases is computed using Euclidean distance [27,28,29,30].

The ML algorithm is applied to a calibration set to perform training of the built models; then, the best model is applied to a validation set (unknown dataset) to test the effectiveness of the classification. The result of the classification prediction is expressed through an error matrix, also known as a consistency confusion matrix. It is a square matrix (n × n), where n is the number of classes in which the columns express the prediction errors and successes, and the lines are the classifiers. The main diagonal lists the samples correctly classified [27,30,31,32,33].

Each spectrum was saved in the “Spectrum” 2.4 software, in JCAMPDX format, then imported into the computer software Pirouette 3.11 for assembling the spectral matrices with 120 lines (samples) and 2619 columns (wavenumber). The Pirouette 3.11 software was used for the application of mathematical methods of noise smoothing and signal preprocessing, as well as for the application of machine learning techniques PCA, HCA, and KNN.

## 3. Results and Discussion

The raw spectra (without pretreatment) in the near-infrared (NIR) region of the intact bacteria, between 14,000 and 4000 cm ^−1^ (700 and 2500 nm), are represented in Figure 2.

It is possible to observe an elevation in the baseline and a very noisy band from 11,000 cm^−1^; in addition, we also have the scattering of the NIR spectra, possibly caused by the lack of homogeneity in the growth of the bacteria. Therefore, it is necessary to use signal preprocessing in order to remove or mitigate spectral noise, baseline elevation and the effect of light mirroring due to diffuse reflectance.

The spectrum obtained is very similar to that already presented in the literature [34], where practically all bacterial samples present almost identical patterns, where a harmonic band of O–H extension at 6900 cm^−1^ and the combined band of the asymmetric O–H vibration is observed and stretched in scissors at 5150 cm^−1^, as already mentioned in other articles the spectral information is quite limited due to the large and wide O-H bands that dominate the spectrum; however, it is known that the structures of the bacterial membrane are combinations of proteins, polysaccharides, nucleic acids and lipids [35].

To reduce the effect of noise, remove redundant information, improve the baseline, reduce the effect of light mirroring due to diffuse reflectance, and obtain the best possible results with machine learning techniques, it was necessary to apply some preprocessing techniques of mathematical signals and methods, such as SNV-standard normal variate, MSC-multiplicative signal correction, EMSC-extended multiplicative signal correction, moving average, Savitzky–Golay, first derivative, second derivative and combinations thereof.

Figure 3 presents the spectra of 120 bacterial samples: *Listeria monocytogenes* (30 samples), *Enterococcus faecalis* (30 samples), *Salmonella enteritidis* (30 samples) and *Escherichia coli* (30 samples), between 4000 and 9900 cm^−1^, after preprocessing using SNV—standard normal variate and smoothing using moving average with 25 windows size.

When a more specific study is carried out on the spectrum after applications of the first and second derivatives, it becomes possible to elucidate some additional bands: at 8670 cm^−1^ attributed to the stretching of the second harmonic C–H, band approximately at 7100 cm^−1^ can be attributed to the O–H of teichoic acid [36], several bands in the region between 4250 cm^−1^ and 4250 cm^−1^ possibly associated with aliphatic C–H groups [37]. At 5620 cm^−1^, the first overtone of CH stretching mode (carbohydrate), bands from 4600 to 4800 cm^−1^ NH, a stretch combination of amide in protein [38].

### 3.1. Unsupervised Machine Learning: PCA and HCA

The matrix used in the PCA and HCA is formed by 120 rows (samples) and 1557 columns (wavenumber).

The dendrogram resulting from the application of the hierarchical cluster analysis—HCA technique that showed the best result was obtained with data treatment centered on the mean, Euclidean metric distance and flexible linkage algorithm as a grouping rule. Figure 4 shows the dendrogram resulting from the application of HCA to NIR spectra. In this figure, it is possible to observe the formation of 4 large clusters: the identification of 4 genera of bacteria, *Listeria monocytogenes* (blue), *Enterococcus faecalis* (gray), *Salmonella enteritidis* (red) and *Escherichia coli*(green). It is also possible to observe the linkage of two large clusters: the classification between bacteria Gram-positive and Gram-negative.

It is possible to corroborate the results obtained through the analysis of clusters by PCA. Figure 5 shows the scores of PC1 vs PC2, which were responsible for describing 86.48% of the total variance, with 79.78% attributed to PC1 and 6.70% to PC2. Through the score chart, we observed that 120 samples of bacteria do not mix; that is, we have 100% identification and authentication of the four types of bacteria: *Listeria monocytogenes* (blue), *Enterococcus faecalis* (gray), *Salmonella enteritidis* (red) and *Escherichia coli* (green). Analyzing PC1, on the right side, we have the group of bacteria Gram-negative, and on the left side of PC1, we have the group of bacteria Gram-positive.

### 3.2. Supervised Machine Learning: KNN

The classification models for bacteria (*Listeria monocytogenes*, *Enterococcus faecalis*, *Salmonella enteritidis* and *Escherichia coli*) were constructed using machine learning algorithms for classification KNN in the NIR spectra studied. For testing the models, 25 were used in each bacteria, totaling 100 spectra for the calibration set and 20 spectra for the external validation set.

The results of the identification of bacteria using NIR spectra are consistent with the results already published in the literature; it is worth mentioning that the selection of specific spectral bands resulted in models that are accurate between 70 to 80%, where samples of a certain species of bacteria were identified 100%, but samples from another species with 70 to 80% identification, so we chose to use the entire spectral region as a “fingerprint”, and in this case, we obtained 100% accuracy for the identification of all bacteria. 

The best model obtained is shown in Table 1, in which the results of the prediction for the identification and classification model created for bacteria (genus and species) can be observed using KNN, as well as their respective predictions of the samples of the calibration set and of the external. The cells that are shaded in gray show the results obtained after the implementation of the model: Bact1(*Enterococcus faecalis*); Bact2 (*Listeria monocytogenes*); Bact3 (*Salmonella enteritidis*) and Bact2 (*Escherichia coli*).

The developed KNN model proved to be efficient for the identification and classification of bacteria (genus and species), as it correctly classified the 100 samples of the calibration set and the 20 samples of the external validation set, that is, 100% accuracy.

The model developed to classify bacteria into Gram-negative and Gram-positive is shown in Table 2, using KNN, 50 samples for Gram-negative and 50 samples Gram-positive do Calibration Set for validation set (10 samples for Gram-negative and 10 samples Gram-positive).

The built supervised machine learning achieved 100% accuracy in classifying the calibration set and external validation set for bacteria Gram-negative and Gram-positive.

The use of machine learning is necessary because when choosing and using a certain wavenumber to recognize a spectral pattern between bacteria and their classification, the model is not as effective, as it becomes very biased towards a pattern (wavenumber), but when we use a with a large spectral range such as the one used, in the form of a “fingerprint”, we are more successful in identifying bacteria and classifying them.

These results demonstrate that the construction of a database using machine learning and NIR spectroscopy can make the identification and classification of bacteria (genus and species) process much faster than traditional methods and much less expensive for companies and research laboratories, as well as for inspection bodies using portable NIRs.

## 4. Conclusions

The Near Infrared spectroscopy using diffuse reflectance, associated with the unsupervised pattern recognition techniques (principal component analysis and hierarchical cluster analysis) and the machine learning techniques (KNN—K-Nearest Neighbor), was able to identify four different types of bacteria, in addition to classifying them into two categories: Gram-positive and Gram-negative. In addition to discriminating quickly, non-destructively and efficiently, this was performed without the use of reagents and the generation of harmful residues, thus helping to preserve the environment. Based on our results, we can determine that our work has great potential to be identification and classification of bacteria, as well as being in line with the world policy of sustainable development and green chemistry.

## Figures and Tables

**Figure 1 sensors-23-07336-f001:**
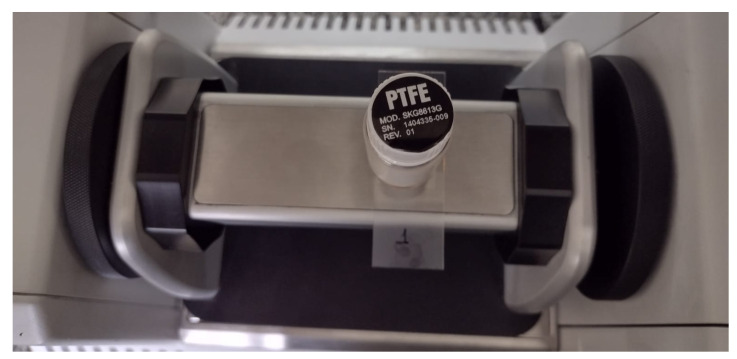
Upper image of the NIR spectrophotometer, containing the slides with bacteria and the reflectance pattern of polytetrafluoroethylene-PTFE of the equipment itself.

**Figure 2 sensors-23-07336-f002:**
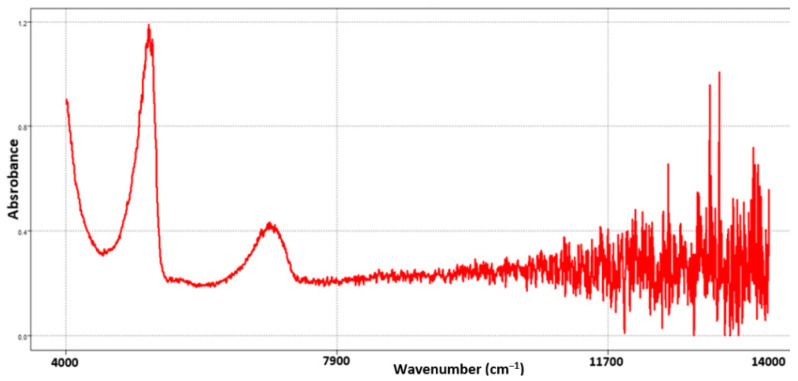
NIR spectra without pretreatment of intact bacteria in the region between 14,000 and 4000 cm^−1^.

**Figure 3 sensors-23-07336-f003:**
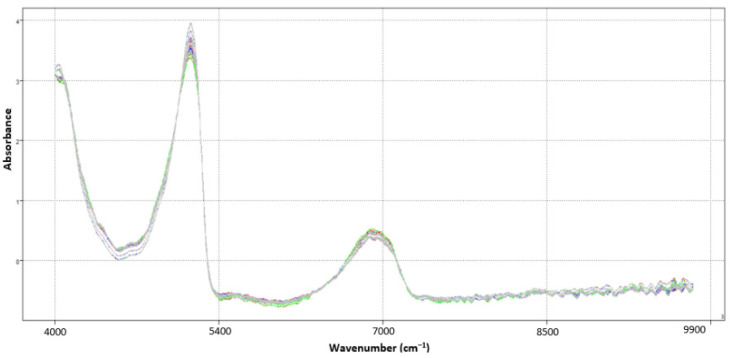
NIR spectra after preprocessing using SNV and smoothing using moving average with 25 windows size, each colored line indicates a sample (bacteria): *Listeria monocytogenes* (30 sample), *Enterococcus faecalis* (30 sample), *Salmonella enteritidis* (30 sample) and *Escherichia coli* (30 sample), between 4000 and 9900 cm^−1^.

**Figure 4 sensors-23-07336-f004:**
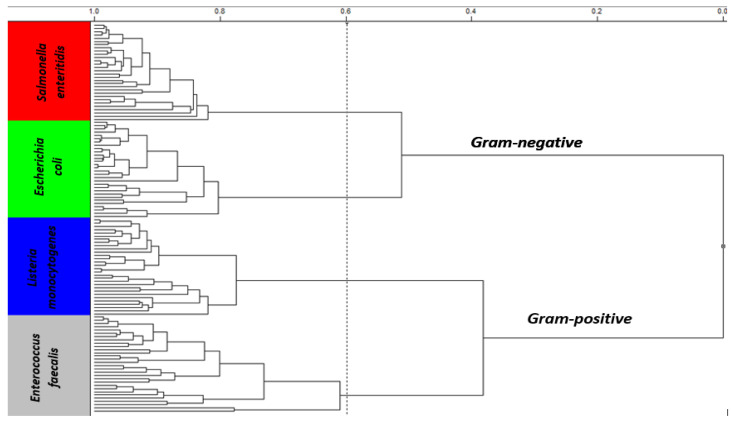
Identification and discrimination of bacteria: *Listeria monocytogenes* (blue), *Enterococcus faecalis* (gray), *Salmonella enteritidis* (red) and *Escherichia coli* (green), obtained by the HCA technique with data centering on the mean, Euclidean metric distance and flexible connection method spectra after preprocessing using SNV and smoothing using moving average with 25 windows size, in the spectral region of NIR between 9900 and 4000 cm^−1^.

**Figure 5 sensors-23-07336-f005:**
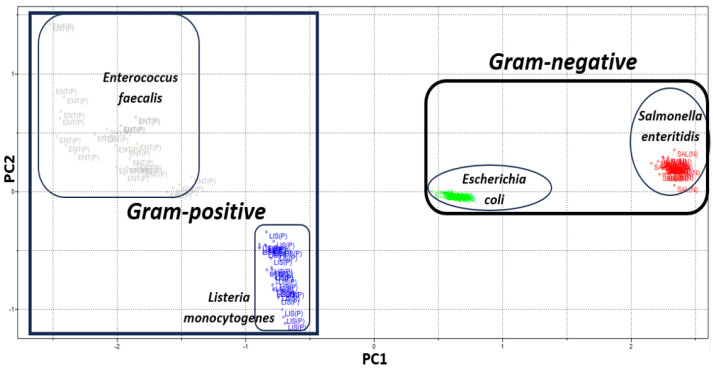
Scores graph PC1 vs PC2 which shows the identification of two large groups of bacteria: Gram-positive (*Listeria monocytogenes* and *Enterococcus faecalis*) and Gram-negative (*Salmonella enteritidis* and *Escherichia coli*), obtained by the PCA technique with data centering on the mean, spectra after preprocessing using SNV and smoothing using moving average with 25 windows size, in the spectral region of NIR between 9900 and 4000 cm^−1^.

**Table 1 sensors-23-07336-t001:** Predictions of the samples (identification of bacteria) used in the creation of the model and also of the samples from the validation set for the NIR spectra using the KNN.

ML KNN	Calibration Set (100 Samples)				Validation Set (20 Samples)			
Bacterium	Bact1	Bact2	Bact3	Bact4	Bact1	Bact2	Bact3	Bact4
Bact1	25	0	0	0	5	0	0	0
Bact2	0	25	0	0	0	5	0	0
Bact3	0	0	25	0	0	0	5	0
Bact4	0	0	0	25	0	0	0	5

**Table 2 sensors-23-07336-t002:** Predictions of the samples (classify bacteria into Gram-negative and Gram-positive) used in the creation of the model and also of the samples from the validation set for the NIR spectra using the KNN.

ML KNN	Calibration Set (100 Samples)		Validation Set (20 Samples)	
Gram	Gram-Negative	Gram-Positive	Gram-Negative	Gram-Positive
Gram-negative	50	0	10	0
Gram-positive	0	50	0	10

## Data Availability

The data presented in this study are available on reasonable request from the corresponding author.

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
