# Peer review of "Rapid and Green Classification Method of Bacteria Using Machine Learning and NIR Spectroscopy"

_sensors, 2023, doi:10.3390/s23177336_

Round 1

Reviewer 1 Report

The manuscript by Farias and coauthors presents a method for classifying bacteria utilizing various machine learning techniques. While machine learning offers a promising approach to classification, there are several areas that require improvement before it could be considered for publication. The following points highlight the areas that need attention:

1.       The introduction section currently suffers from excessive historical information and lacks a clear focus on the motivation behind the research. To enhance the manuscript, the authors should consider providing a more concise and relevant introduction that precisely outlines the problem being addressed and the significance of their work in the context of green chemistry.

2.        One major drawback of the manuscript is the absence of crucial details on how the machine learning techniques are implemented. Readers need to know about the specific methodologies, algorithms, and parameters used to replicate and validate the results. Additionally, more comprehensive information about the datasets used, including their sources, size, and characteristics, is necessary to assess the validity and generalizability of the findings.

The English language in the manuscript needs substantial improvement. There are numerous instances of incomplete sentences and grammatical errors that hamper the readability and overall comprehension of the text. A thorough proofreading and editing process is necessary to ensure a clear and coherent presentation of the research.

Author Response

The requested corrections were made, the introduction was reduced, paragraphs and figures were added that better described the methodology, the software used and the spectral treatment.
The text added to the article is in blue color. The  text underwent a new grammatical review.

We are at your disposal for further information.

Reviewer 2 Report

sensors-2508318

The manuscript (communication) entitled “Rapid and Green Method classification of bacteria using Machine Learning and spectroscopy NIR” is a well written comprehensive piece of work and in line with the demand of prevailing environmental conditions. However following changes /improvement are needed before publication:

The title needs some reshuffling of words (Method, NIR) and should be modified to “Rapid and Green classification Method of bacteria using Machine Learning and NIR spectroscopy”.

1. Abstract and Introduction:

There is no need to write make model of NIR in abstract as it is described in Materials and Methods section.

Line-34-35: The word environment is written twice.

In my opinion, the Introduction section is overwritten (3 pages) and should be reduced to 2 pages by removing some unnecessary part (like from line 69-84).

Some abbreviations like SIMCA (line 117), AI (line 197), SNV (line 214) need to be in full form at their first use as these are not commonly used ones. Moreover, both pattern types i.e. – and ( ) used for abbreviation purpose, please stick to one pattern.

2. Materials and Methods:

Line 169: I think for should be replaced with four.

Line 179-180: What is PFTE used as reference material?

3. Results and Discussion:

Figure 1: What does the colour represents in the graph? If they represent bacteria then please improve your graphical representation. Moreover, all samples have nearly same absorbance. Give rationales for same absorbance.

Why 30 samples were used. Not 50 or 20? and also there is no description of type of samples used, means broth or slant or ampule.

Table 1 and 2: Both tables are without legends. So please add table legends on both tables for easy understanding of table contents.

Line 292: Please see and remove line 292 in Authors contribution statement.

Line 307-312: Have a relook.

Moreover, funding is repeated twice (in funding and acknowledgement)

4. References

Please make sure all references are as per journal guidelines.

Author Response

The requested corrections were made, the title was changed as suggested.
The abstract has been rewritten.
the intro has been shortened.
Paragraphs and figures were added to better describe the methodology, a figure with a sample on the slide was added.
Information about the software used and the spectral treatment has also been added.
The text added to the article is in blue color.
The text underwent a new grammatical review.
We are at your disposal for further information.

Reviewer 3 Report

This communication reports on the development of a rapid and green method for the classification of bacteria using machine learning and Near Infra-Red spectroscopy. The work reported is important form environmental point of view. Both experimental and machine learning methods have been used to develop identification and classification models of bacteria. The communication is a good written scientific document and publishable in Sensors after modification/revision in the light of following comments.

Abstract:

1.       Delete  “using a BOMEM MD- 1600 spectrophotometer equipped with a diffuse reflectance accessory, with a resolution of 8 cm-1,  in the spectral region from 14,000 to 4,000 cm-1”. It should only be written in methodology.

2.       They write “Gram-positive and   Gram-negative, obtaining 100% accurate results”. Is it possible to get 100% accurate results?

Introduction:

3.       They write “Among the 3 existing spectroscopy techniques in the infrared region, near-infrared 

spectroscopy has stood out for having a lower price and portable and low-cost equipment   when compared to the others  (MID and FAR)”. I think NIR spectrophotometers are very expensive as compared to MIR spectrophotometers. For example please check price list of Shimadzu corporation products.

4.       they write “Transmittance mode is used usually for liquids or gases samples, where the samples 

are placed in glass/quartz cuvettes  with typical optical paths varying from 1 to 50 mm  and  NIR transmittance is measured on the other, results can be expressed in transmittance or absorbance, depending on the brand of each NIR equipment”.  Transmittance mode also works very well with solid samples.

5.       Highlight novelty of this work by writing comprehensive problem statement at end of introduction.

Materials and methods:

6.       Please write complete name of PFTE.

7.       Write about sampling method of 120 bacteria samples.

Results and Discussion:

8.       There is no interpretation of bands observed in Figure 1.

9.       How can the authors correlate the findings of ML with NIR results?

10.   They need to compare their results with literature.

11.   More discussion in the light of updated literature will make the communication more attractive for readers.

12.   There are several typo/grammatical mistakes in the written draft. correction is suggested.

Moderate editing of English language required

Author Response

The requested corrections have been made.
The abstract has been rewritten.
The intro has been shortened.
Paragraphs and figures were added to better describe the methodology,
information about the software used and the spectral treatment was also added.
Added paragraphs about the spectral interpretation of bacteria and the existing literature, as well as the reason for the use and necessity of machine learning for this work.
The text added to the article is in blue color.
The text underwent a new grammatical review.
We are available for further information.

Round 2

Reviewer 1 Report

The manuscript has been greatly improved and most of my previous comments have been addressed. However, the introduction part could be more concise.

Reviewer 3 Report

The revised version is much better as compared to the the orignal submission and publishable in Sensors

Minor editing of English language is required